# Towards a Postgraduate Oncology Training Model for Family Medicine: Mixed Methods Evaluation of a Breast Oncology Rotation

Michelle B. Nadler [1,2,*], Brooke E. Hofbauer [2], Melinda Wu [3,4], Susan Hum [4], Christine Elser [1,5] and Joyce Nyhof-Young [3,4]

1    Division of Medical Oncology & Hematology, Department of Medicine, University of Toronto, Toronto, ON M5T 2S8, Canada
2    Princess Margaret Cancer Centre, University Health Network, Toronto, ON M5G 2C4, Canada
3    Department of Family & Community Medicine, University of Toronto, Toronto, ON M5T 2S8, Canada
4    Women's College Hospital Family Practice Health Centre, Toronto, ON M5S 1B2, Canada
5    Mount Sinai Hospital, Toronto, ON M5G 1X5, Canada
*    Correspondence: michelle.nadler@uhn.ca

**Abstract:** Background: Family physicians have low knowledge and preparedness to manage patients with cancer. A breast oncology clinical rotation was developed for family medicine residents to address this gap in medical education. Objectives and Methods: A breast oncology rotation for family residents was evaluated using a pre-post knowledge questionnaire and semi-structured interviews comparing rotation (RRs) versus non-rotation (NRRs) residents. Quantitative and qualitative data were collected via a pre-post knowledge questionnaire and semi-structured interviews, respectively. Analysis: Quantitative data were analysed using descriptive statistics and paired t-tests to compare pre-post-rotation knowledge and preparedness. Qualitative data were coded inductively, analysed, and grouped into categories and themes. Data sets were integrated. Results: The study was terminated early due to the COVID-19 pandemic. Six RRs completed the study; 19 and 2 NRRs completed the quantitative and qualitative portions, respectively. RRs' knowledge scores did not improve, but there was a non-significant increase in preparedness (5.3 to 8.4, $p = 0.17$) post-rotation. RRs described important rotation outcomes: knowledge of the patient work-up, referral process, and patient treatment trajectory; skills in risk assessment, clinical examination, and empathy, and comfort in counseling. Discussion and Conclusion: Important educational outcomes were obtained despite no change in knowledge scores. This rotation can be adapted to other training programs including an oncology primer to enable trainee integration of new information.

**Keywords:** family medicine residents; medical education; breast oncology; breast cancer; rotation evaluation; program evaluation; mixed methods; knowledge; skills; communication; survivorship

## 1. Introduction

Breast cancer (BC) is the most common malignancy diagnosed in women, and there is an increased prevalence of women living with side effects of cancer and its treatment [1]. As a result, family physicians' (FPs) responsibilities for patients with cancer have expanded to include screening, surveillance, survivorship, and palliative care [2]. Care complexity has also increased, with specific knowledge required to know the criteria for genetic testing (e.g., coverage for 19-susceptibility gene panel testing in Ontario if lifetime risk > 10%), high-risk screening, surgical/reconstruction options, and long-term survivorship issues [3,4]. FPs require proficiency in the CanMEDS "communicator" and "collaborator" roles [5] to collaborate effectively with the oncology team.

Despite this, oncology education is under-emphasised in medical school and postgraduate family medicine (FM) training, and oncology teaching is ranked poorly by

trainees [5–7]. The CanMEDS framework highlights that proficiency in the communicator and collaborator roles is best gained through direct clinical exposure [8,9], but oncology curricula provide trainees with few practical applications to acquire these skills [2,10–13]. Canadian FPs and FM residents report a lack of knowledge, comfort, and preparedness to manage cancer care in practice [6], and few FM programs have survivorship curricula [14]. Likewise, oncology education for American FPs has also been shown to be inadequate, with less than a quarter feeling confident identifying cancer recurrence and potential long-term effects of cancer treatment [15]. Inadequate training in cancer survivorship care and low awareness of survivorship issues could lead to missed or delayed diagnoses and poorer health outcomes in these patients.

A pilot breast oncology rotation was designed in 2013 for Women's College Hospital (WCH) FM residents to attend at the Princess Margaret Cancer Centre (PMCC) in Toronto, Canada. Based on data supporting short clinical rotations as effective interventions [11,16,17], residents attend 7–8 weekly half-day outpatient clinics (during a 2-week period) including genetics, breast diagnostics, surgical oncology, medical oncology, and survivorship/treatment transition (Supplementary Table S1). Approximately 15–20 residents participate yearly. Given the demand for this clinical rotation, a program evaluation was undertaken to assess rotation effectiveness prior to expansion into other educational sites. Our aim was to evaluate (a) rotation impact on residents' breast oncology knowledge, perceived practice preparedness, and remaining educational needs, and (b) rotation components important in achieving these outcomes. We compared FM residents participating in the rotation to those near the end of their residency who had not participated. Results can inform PMCC/WCH rotation improvements and determine whether this model is adaptable to other local, national, or international educational sites.

## 2. Methods

### 2.1. Design, Setting, and Participants

The program evaluation was a pragmatic mixed-methods study [18,19] comparing University of Toronto FM residents who rotated through the PMCC/WCH rotation ("rotation residents" (RRs)) to FM residents who did not ("non-rotation residents" (NRRs)). RRs were recruited on a rolling basis as they undertook the rotation (only 7 became eligible due to early study termination due to COVID-19); they were contacted 1 week prior to starting the rotation and sent one reminder email. Approximately 150 eligible NRRs were recruited based on an e-mail invitation (with two reminders) to participate in the study. Sample size was calculated assuming an 18-month recruitment period. For RRs, we expected 60% participation (of 15–20/year), a total of 9–12 residents. For NRRs, we expected a 20–30% (of approximately 160 s year FM residents per year) response rate, yielding 32–48 participants.

We used quantitative data to measure objectively residents' knowledge and qualitative data to understand their reasons for choosing the elective and their experiences during the rotation. Quantitative data were collected via a knowledge questionnaire delivered to the RRs at baseline just prior to and post rotation (Supplementary Figure S1). The same baseline questionnaire was delivered to NRRs during the last 6 months of residency training. Qualitative evaluation was conducted via semi-structured interviews at the end of the clinical rotation (RRs) or after questionnaire completion (NRRs). Participating residents provided signed informed consent and received a $10 gift card for each study component completed. This study was approved by the University of Toronto Research Ethics Board (UT REB #38224).

### 2.2. Data Collection, Questionnaire, and Interview Guide

The on-line questionnaire was developed based on rotation objectives, a literature review, and iterative input from clinician-teachers, medical educators, and an education scientist. It comprised 26 questions: 8 demographics, 17 multiple-choice knowledge questions, and 1 (with 11 sub-sections) assessing preparedness using a 5-point Likert scale. The post-rotation questionnaire was similar, except demographic questions were removed

and 6 questions about the rotation were added. The surveys were pilot tested by three FM residents for face and content validity and delivered. A modified Dillman approach was used to recruit participants, with an email to eligible residents and generic reminder emails at two-week intervals [20]. Survey data were collected online using Qualtrics online software. The semi-structured interview guide was developed iteratively by project collaborators and pilot tested with a former RR and NRR. It focused on prior oncology experience, the FP role in managing oncology care, residents' practice preparedness, and a final question about breast cancer screening. For RRs, there were additional questions about reasons for rotation participation and overall experience. For NRRs, there were additional questions about rotation awareness, adaptability, and perceived educational needs. Interviews were conducted via telephone. The RR interviews were conducted by MBN, a breast medical oncology fellow with training in education and qualitative interviewing. The NRR interviews were conducted by BEH, a family medicine resident who had previously completed the rotation.

### 2.3. Analysis

Questionnaire data were downloaded from Qualtrics as an SPSS file and descriptive statistical analyses (frequencies and percentages for categorical variables; means and standard deviations for continuous variables) conducted using IBM SPSS Statistics version 25.0 (IBM Corp, Armonk, NY, USA). Points were assigned and totaled for correct answers to the 17 multiple choice questions. Preparedness Likert-scale items were dichotomised to un-prepared [1–3] and prepared [4,5]. Paired t-tests compared residents' knowledge and preparedness scores pre- and post-rotation, and unpaired t-tests compared average scores between rotation and non-rotation residents. A $p$ value < 0.05 was considered significant. A descriptive, inductive approach was taken for the qualitative analysis, as previously described by Percy and Kostere [21]. Qualitative interviews were digitally recorded, transcribed verbatim, and de-identified. First, two study-team members (MNB and BEH) read the initial transcripts completely multiple times and independently assigned inductive codes to sections of text using terms from the transcripts themselves. Next, results were compared between coders and a codebook made to define the codes and provide examples. This codebook was used to deductively code the next few transcripts, using a consensus approach to resolve discrepancies [21,22]. If either coder felt that a piece of text was relevant to the research question but did not fit any of the codes in the codebook, this was discussed amongst the team and a new code was established and added to the codebook. Early transcripts were re-reviewed to look for other instances of new codes. The team participated in further interpretive analysis to group the codes into categories and higher-level themes. Saturation was pragmatically defined as occurring when further interviews did not lead to new codes, themes, or enrich understanding of prior codes and themes. Qualitative and quantitative RRs data analyses were integrated with each other and corresponding NRRs analyses to enhance understanding of residents' educational contexts and needs [22,23]. This integration was done by comparing important codes and categories (e.g., 'survivorship' or 'knowledge') to the respective quantitative questions addressing these issues. The team assessed corroboration of the qualitative and quantitative results; if the qualitative results did not support the quantitative ones, the team hypothesised reasons for the discrepancies.

## 3. Results

### 3.1. Demographics

In March 2020, the rotation was stopped due to the COVID-19 pandemic as residents were re-deployed. The study was therefore terminated early. Six of seven eligible RRs (86%) completed baseline questionnaires and five (71%) completed the full study. Of NRRs, 28/150 (19%) participated in the online questionnaire, but only 19 (13%) completed any knowledge questions. Two NRRs participated in interviews. Most participants were women (71% RRs and 79% NRRs), with an average age of 28–29. See Table 1 for participant characteristics.

**Table 1.** Demographic Information of Participating Residents.

|  | Rotation Residents (*n* = 7) | Non-Rotation Residents (*n* = 28) |
|---|---|---|
| Age, mean (range) | 29 (27–33) | 28 (25–32) |
| Women, *n* (%) | 5 (71.4) | 19 (67.9) |
| Anticipated Career Plan, *n* (%) | | |
| - Community-Rural | 0 | 7 (25) |
| - Community-Urban | 3 (42.9) | 10 (35.7) |
| - Academic | 2 (28.6) | 5 (17.8) |
| - Academic with specialisation in women's health | 1 (14.3) | 1 (3.6) |
| - Other/Missing | 1 (14.3) | 5 (17.8) |
| Didactic Oncology Exposure/Experience in medical school, including lectures and case-based learning: *n* (%) | | |
| - Few (1–5) | 4 (57.1) | 15 (53.6) |
| - Moderate (6–10) | 1 (14.3) | 8 (28.6) |
| - In depth (>10) | 1 (14.3) | 1 (3.6) |
| - Missing | 1 (14.3) | 4 (14.3) |
| Didactic Oncology Exposure/Experience in residency, including lectures and case-based learning: *n* (%) | | |
| - None | 0 | 7 (25) |
| - Few (1–5) | 5 (71.4) | 16 (57.1) |
| - Moderate (6–10) | 1 (14.3) | 1 (3.6) |
| - In depth (>10) | 0 | 0 |
| - Missing | 1 (14.3) | 4 (14.3) |
| Estimated number of patients with active cancer or a history of cancer seen in residency so far: | | |
| - 1–10 | 2 (28.6) | 7 (25) |
| - 11–25 | 3 (42.9) | 10 (35.7) |
| - 26–50 | 1 (14.3) | 2 (7.1) |
| - More than 50 | 0 | 5 (17.9) |
| - Missing | 1 (14.3) | 4 (14.3) |

### 3.2. Quantitative Results

On average, RRs scored $10.17 \pm 1.47$ out of 17 on the knowledge questionnaire at baseline and $11.4 \pm 1.67$ post-rotation (not significant (ns)). NRRs scored lower than RRs at baseline ($8.13 \pm 3.24$, ns; $p < 0.15$). RRs improved their scores on endocrine therapy questions (efficacy and side effects) from 60% to 100% and 0% to 60% respectively, and questions about surveillance for average-risk women improved from 16% to 80% (Supplementary Table S2). At baseline, RRs felt more prepared to manage oncology issues in practice compared with NRRs (3.16 vs. 5.3, mean difference $2.14 \pm 1.4$, $p = 0.14$). Rotation participation resulted in a non-significant preparedness increase (5.3 vs. 8.4, mean difference $3.1 \pm 2.1$, $p = 0.17$).

### 3.3. Qualitative Results

There were three major interview themes: (i) rotation outcomes, (ii) rotation structure and content, and (iii) resources and recommendations, as summarised in Tables 2 and 3 and Supplementary Tables S3 and S4. Despite early study termination, data saturation was reached for the RRs, but not for NRRs as only two interviews were conducted.

**Table 2.** Supportive resident quotes for theme of rotation outcomes. Additional quotes in Supplementary Table S3.

| Theme: Rotation Outcomes | | |
|---|---|---|
| Category: Knowledge | Codes: 1.Foundation/Framework 2. Referral Process and (Care) Trajectory 3. Roles 4. Management 5. Surveillance/Survivorship | Rotation Residents: 004: Having this experience gives me more clarity about their [the patient] experience and the multiple different specialists they have to see . . . what the surgery was like . . . what that chemotherapy regimen looked like... later on if they're on any hormonal therapy, having a better understanding of how to counsel them around it, talk about potential side effects, and how to manage them long-term when they're on it for years. 002: Often the radiologist would say based on the fact that this is a cellular fibroadenoma, refer to surgery. And I think if I had not done this rotation, I would be like, 'Oh, fibroadenoma, that's perfectly fine.' . . . but it would not have triggered an internal dialogue, to be like, 'Maybe this is a phyllodes tumour.' Non-Rotation Residents: 101: Post treatment screening people that I might have a bit more of a challenge in doing . . . myself, I wouldn't know the guidelines off the top of my head. |
| Category: Skills | Codes: 1.Risk Assessment 2. Information Synthesis/Application 3. Physical Exam 4.Empathy | Rotation Residents: 002: A lot of my takeaway was learning about how risk gets stratified and the tools that oncologists and surgeons have at their disposal. The IBIS calculator actually is really cool, and you can plot where people are in real time according to their risk. 004: Now when I see reports, or even one of the imaging reports, I'm able to better process it and not feel, I guess scared or nervous . . . 004: Getting to do a bunch of breast exams was definitely great . . . Learning about how to feel the axillary lymph nodes properly was really helpful, and actually to feel lots of breast masses was really helpful as well. 004: In a way, I think being more empathetic to what they're going through . . . |
| Category: Comfort (Counseling) | Codes: 1.Counseling 2. Collaboration 3. General Comfort | Rotation Residents: 009: [Patients] often ask me for advice on what to do, and I just felt like I could speak from a more knowledgeable place about a very common condition and what generally happens to women who are diagnosed with breast cancer. Non-rotation residents: 103: When it comes to managing patients after they've maybe entered that system and there's more nuanced aspects to their care, that's something that I don't feel as comfortable with [or] managing patients that have ongoing treatment, whether it's radiation or chemotherapy, because I'm not entirely sure of what special considerations to keep in mind. |
| Category: Transferability | | Rotation Residents: 003: I still have an appreciation for the principles of therapy, which could be surgery, chemo, or rads, but I don't know really how much I could apply that much more in detail for other cancers. 009: Certainly I felt a lot more comfortable, as sad as it sounds, with delivering bad news and I'm sure that translates to almost all cancers. |

**Table 3.** Supportive Resident Quotes for Theme of Rotation Structure and Content.

| Theme: Rotation Structure and Content | | |
|---|---|---|
| Category: Clinics | Codes:<br>1. Variety<br>2. Service<br>3. Breakdown | <u>Rotation Residents:</u><br>004: I really liked the fact that I got exposed to a variety of different clinics. I was in an After Cancer Care Clinic and then med-onc and surgical oncology clinics. I felt that gave me good diversity in what each specialty would do, and what my patients would go through when they see each specialist. . . . All of the clinics really supplemented each other and gave me a holistic or comprehensive perspective of the care that patients would get.<br>003: I worked with med-onc, surg-onc, and the GP oncologist so all three, and even one day with a geneticist, a genetic counsellor. I really liked that because I saw similar patients from some different perspective. Each professional had their own lens.<br>003: The other thing which might be interesting... would be the Rapid Diagnostic Clinic, and I see them after they've gone to have their biopsy and everything done. I remember one of the doctors explaining to a patient even about the breast MRI, that you have to be on your stomach, and I never would have thought to counsel on these things. |
| Category: Exposure | Codes:<br>1. Outside rotation<br>2. Clinicians<br>3. Patients | <u>Rotation Residents:</u><br>000: I found that I hadn't had much exposure to oncology or breast care before the rotation, and I found that I got a lot of exposure to both fields. I got good support, good teaching in general, and just saw a lot of mostly oncology cases that I probably would have never seen and then wouldn't have had exposure to otherwise; so I found it really helpful.<br>004: I didn't really find I had any really good formal teaching about how do you work something up, how do you investigate further, and the referral pathway, who is the best person to refer to? I didn't really get a sense... I may have encountered a few patients in family medicine who have gone through it and worked with preceptors that way.<br>003: Another super-key point is just the volume of breast exams that I did during my two weeks. . . . I got to see so much variation of normal and also a little bit of abnormal.<br><u>Non-Rotation Residents:</u><br>103: I feel like the training makes me feel comfortable with screening and the referring on something that is abnormal, but I think the nuances of what happens with someone who has already been treated or has a specific condition are not so clear to me because we don't really have any formal training on that. |
| Category: Teaching | Codes:<br>1. Didactic<br>2. Clinical skills | <u>Rotation Residents:</u><br>101: Didactic teaching focused on screening and post cancer follow-up would be helpful.<br>002: [The staff took] a lot of time to even discuss the patient care aspect in terms of how you should communicate with the patients and nuances of doing a physical exam that were super high yield. She got me to palpate abnormalities. She went over the screening guidelines, which she knew those very well. And I should know well. That was super high yield.<br>004: Yes, one thing I really appreciated was most of the staff would cater to what I might be seeing in my clinic and give me little tidbits and little pearls here and there . . . The focus was always the big picture of common side effects patients may present with and that kind of stuff, and I found that really helpful.<br>009: the preceptor that I was with really dedicated themselves to teaching. Often we would spend a little bit of time after clinic discussing different cases and how to approach them. I think he also spent some time teaching me about benign breast disease which I didn't have a lot of exposure to, which I wished I had more of. |

### 3.4. Theme 1-Rotation Outcomes

The most robust theme, rotation outcomes, included the categories of knowledge, skills, comfort related to management and counseling, and knowledge transferability to non-breast oncology practice (Table 2). Residents from both cohorts reported limited prior foundational oncology knowledge. RRs reported increases in knowledge regarding the work-up and referral process for breast concerns, treatment, and trajectories of patients with breast cancer. Both groups emphasised these as critical in delivering patient care. Throughout the rotation, residents learned about the roles of each type of oncologist. In doing so, they were better able to appreciate the role of family physicians in screening, diagnosis, collaboration, communication, and advocacy. Both resident groups felt that their residency training prepared them well for average-risk breast cancer screening, but not for managing common oncologic concerns, post-treatment surveillance, and survivorship issues.

RRs stated that they acquired important skills, including risk assessment, clinical examination, information synthesis/application, and empathy. Specific skills important to FPs included breast cancer risk assessment and breast examination. RRs also learned to better synthesise information, for example, the nuanced area of breast cancer screening in women ages 40–49. They expressed that they had gained more empathy and were better able to relate to patients with breast cancer. While NRRs described empathy, it was limited to a personal experience with cancer or professional experience with palliative care.

RRs described being better able to anticipate the trajectory of care of newly diagnosed patients, allowing them to feel more comfortable counseling patients. They also expressed increased comfort collaborating with oncology specialists and were able to identify the correct service for referral and inquiries. Both cohorts said they were comfortable with breaking bad news to a patient, a skill emphasised in FM training and developed throughout other mandatory rotations. Finally, RRs expressed mixed opinions as to whether knowledge, skills, and comfort gained from the rotation were transferable to other common cancers. Some noted that it was difficult to know without completing a non-breast oncology rotation. Others felt that select foundational concepts, such as patient communication would be transferable.

### 3.5. Theme 2-Rotation Structure and Content

This theme included three categories: clinics, exposure, and teaching (Table 3). RRs stated that the 2-week duration was sufficient to meet learning objectives given ample learning opportunities in the diverse and high-volume clinics. Clinic variety provided a comprehensive knowledge base about patients' experiences from diagnosis through treatment and survivorship; however, the learning curve was steep. Both resident groups said they had limited prior clinical oncology exposure. NRRs spoke about prior oncology experience in palliative care rotations, but this was limited to end-of-life care. Neither resident group had foundational knowledge in oncology management. Both discussed the importance of exposure to diverse patient presentations and specialists to increase their knowledge and comfort. RRs said learning was enhanced by informal case discussions and observing staff, particularly when they discussed new diagnoses or management of treatment non-adherence. They noted the excellent teaching provided via one-on-one staff interactions and case-based discussions, although some identified a lack of didactic, family practice-focused teaching.

### 3.6. Theme 3-Rotation Resources and Recommendations

Recommendations for rotation improvement included adding an oncology primer and rotation promotion (Supplementary Table S4). All residents strongly desired a primer, or resource to provide key concept review prior to the rotation. They stated that this would help them to integrate new information during the initial steep learning curve. Recommended content for the primer included key aspects of work-up and diagnosis, surgical options, common chemotherapeutic regimens, indications for radiation, and differences

between endocrine therapies. Suggested formats were a printable handout, PowerPoint presentation, and online modules.

With respect to scheduling, RRs appreciated the variety in clinics, but requested that some clinics be arranged according to their individual learning objectives and career goals. For example, residents unable to work with the genetic counselors felt they could have benefitted from additional training in breast cancer risk assessment. Given that NRRs were unaware of this rotation, they suggested advertising it amongst family medicine trainees.

## 4. Integration and Discussion

### 4.1. Summary of Findings

We present a concurrent mixed-methods evaluation of a breast oncology selective for FM residents. Quantitative results showed no change in RRs knowledge post-rotation (possibly due to low power); however, our qualitative results demonstrated increases in knowledge and other important outcomes. RRs reported improvements in risk assessment and physical examination skills, a better understanding of specialist roles and a patient's trajectory through oncology care, enhanced ability to empathise with patients, and increased comfort counseling patients and collaborating with oncology team members. Both RRs and NRRs lacked foundational knowledge in oncology, making it difficult to integrate new information.

### 4.2. Knowledge

While quantitative data show no significant improvement in content knowledge scores, RRs described other important increases in knowledge and skills related to trajectory, management, and communication. This discrepancy may exist due to their reported lack of general oncology knowledge prior to the rotation. Without background, incorporating new information in a short rotation can be difficult. This lack of foundational knowledge has also been reported in the literature [11,12,16], but not its impact on a trainee's performance in a rotation. Therefore, residents may find it difficult to differentiate between standard of care delivery versus when they observe an oncologist deviate due to a patient's individualised circumstances. A basic oncology primer provided prior to the rotation and more didactic teaching throughout may enable residents to integrate new information and improve content knowledge. This also suggests to us that medical curricula should provide more basic oncology training, focused more on clinical management and less on pathology.

### 4.3. Non-Knowledge Rotation Outcomes

RRs described important rotation benefits (as above) and a non-statistical improvement in preparedness score, with a moderate effect size, notable even with low power. Patients often turn to their FP for advice regarding cancer management, but FPs report low knowledge and comfort with such discussions [24]. RRs felt they could better support patients based on knowledge gained during the rotation. Further, the majority of primary care providers reported that they are expected to manage the psychological consequences of cancer diagnosis and treatment, but only a quarter of this sample felt adequately prepared to do so [25]. Our study showed that this 2-week rotation helped FM residents develop increased empathy and understanding of the patient experience, which allowed them to better connect with and support their patients. This may also explain why preparedness scores increased despite no overall change in content knowledge. Two recently reported curricula developed for medical students and internal medicine residents [26,27] successfully improved understanding and comfort with cancer survivorship but did not describe the additional benefits reported in our study. Our results imply that educational opportunities for primary care physicians should consider both content knowledge and outcomes related to their role in cancer care, care trajectory, and survivorship.

### 4.4. Surveillance and Survivorship

In both quantitative and qualitative results, an important positive outcome was FM increased knowledge and comfort in dealing with issues in the surveillance and survivorship periods. This was evidenced by an improvement in survey questions related to endocrine therapy and post-diagnosis surveillance and reported in interviews. A related outcome gleaned only from interviews was that of increased comfort collaborating with the oncology team during these periods. This is particularly important given FP reports of low confidence in evaluating, managing and collaborating with the oncology team regarding late effects of treatment [28,29]. An Australian training program in which primary care providers complete short observational placements in a tertiary oncology centre also showed increased participant knowledge regarding survivorship care and improved clinical relationships with specialist teams [30]. We recommend that a focus on acute and long-term side effects of systemic therapies, and routine surveillance guidelines should be incorporated into rotations designed for FM residents.

### 4.5. Transferability to Common Non-Breast Cancers

The transferability of knowledge and skills to other common cancers was only evaluated qualitatively. Most residents suggested that some basic oncologic concepts and communication skills were transferable. If the goal of a rotation is to provide FM residents with knowledge and skills transferable to other cancers, then the rotation should be adapted or broadened to include clinics with other common cancers. This new rotation could be a "general oncology" rotation and the more specific breast-oncology rotation could then be reserved for residents with specific interest in breast oncology or women's health (as examples). Having two distinct oncology rotations may attract a more diverse set of oncology focused residents and increase resident access to such rotations at our university. Although we did not specifically assess the international generalisability of our results, we feel our results maintain applicability outside Canada, given similar contexts (the under-representation of oncology in undergraduate medical education, the multi-disciplinarity of oncology, and the importance of clinical exposure) [11,15]. Our breast oncology selective and other medical oncology training programs may be adapted to other countries to help address the problem of insufficient workforce to provide cancer care [31].

### 4.6. Limitations

This study has limitations. The sample size was small given the early termination due to the COVID-19 pandemic. Therefore, the value of the quantitative data on content knowledge and preparedness was limited due to low power. This also limits the ability to truly integrate the findings in a robust mixed method analysis. However, the RR interviews provided rich explanatory data to integrate with the quantitative findings, and we gained important insight into the rotation from the results of the quantitative evaluation that we would not have if it were omitted. We were only able to interview two NRRs, so data saturation was not reached for this group, but they were not the main study focus. Second, our novel knowledge questionnaire lacked both internal and construct validity as it had not been previously used. However, we confirmed its face and content validity through extensive team discussion, review with content experts, and piloting with a resident. Third, our baseline data indicated that the RRs are different from the NRRs, with a greater interest in women's health and higher baseline knowledge. They were likely more engaged with the rotation material; it is unknown how the rotation might impact residents without these interests. Nonetheless, both RRs and NRRs described a lack of an oncology foundational knowledge and low comfort caring for these patients; therefore, this rotation (or a 'general oncology' rotation) would likely be beneficial.

Despite these limitations, we demonstrated that a brief 2-week breast oncology rotation for FM residents can improve knowledge of surveillance and survivorship, individualised risk assessment, empathy, and increased comfort in collaboration and counseling. The following components were seen as important to achieve these outcomes: high-volume

clinics allowing physical examinations (including palpation of abnormal findings), direct teaching and feedback relevant to a FP, and a variety of specialty clinics with the opportunity for tailored scheduling to address resident interests and educational needs. Our study identified opportunities to improve the rotation with the provision of a primer for residents prior to the rotation. This would include key oncology terms, indications for each modality of treatment, basic management guidelines, endocrine therapy side effects, surveillance guidelines, and common issues presenting in the survivorship period.

## 5. Conclusions

The PMCC/WCH rotation positively impacted residents' knowledge and skills in important areas of breast oncology and increased their sense of preparedness for patient care but did not improve overall content knowledge. These outcomes were accomplished through exposure to a diverse range of high-volume clinics, medical specialties, and dedicated teachers. Providing residents with a primer prior to the rotation and allowing residents to choose elective clinic options can further improve the rotation. If the goal of the rotation is to achieve these outcomes for other common cancers, then these clinics should be included within a distinct "general oncology" rotation. Our breast oncology rotation can serve as an adaptable model at educational institutions where such resources are available.

**Supplementary Materials:** The following supporting information can be downloaded at: https://www.mdpi.com/article/10.3390/curroncol29090510/s1, Figure S1: east Selective Template Schedule; Table S1: Quantitative Results; Table S2: Quantitative Results; Table S3: Supportive Resident Quotes for Theme of Rotation Outcomes; Table S4: Supportive Resident Quotes for Theme of Resources and Recommendations.

**Author Contributions:** Conceptualization M.B.N., M.W., C.E., J.N.-Y. Data collection M.B.N., B.E.H. Analysis M.B.N., B.E.H., C.E., S.H., J.N.-Y. Resources C.E., M.W., J.N.-Y. Writing—original draft preparation M.B.N., B.E.H., S.H. Writing—review and editing M.B.N., C.E., J.N.-Y. All authors have read and agreed to the published version of the manuscript.

**Funding:** M.B.N.'s fellowship was supported as a Dream Hold'Em For Life foundation and the Jim Nicol Foundation. Christine Elser receives research funding from the Jim Nicol Foundation.

**Institutional Review Board Statement:** This study was approved by the University of Toronto Research Ethics Board (UT REB #38224).

**Informed Consent Statement:** Informed consent was obtained from all subjects involved in the study.

**Data Availability Statement:** Primary data is stored securely and available upon request.

**Conflicts of Interest:** The authors declare no conflict of interest.

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
