# Peer review of "Towards a Postgraduate Oncology Training Model for Family Medicine: Mixed Methods Evaluation of a Breast Oncology Rotation"

_curroncol, doi:10.3390/curroncol29090510_

Round 1

Reviewer 1 Report

Thank you for giving me to review your manuscript. This manuscript is interesting and scientifically meaningful for considering Breast Oncology Rotation for Family Medicine Residents. Regarding the contents, the following revision should be considered.

The title should contain the significance of this study.

The abstract should contain the meaningfulness of this study.

In the background, the authors should follow the rationale of paragraph writing.

In the introduction, the researchers should show the research question clearly.

This study should describe why this study used mixed method approaches to investigate the research question clearly.

The sample calculation should be described in the analysis part.

The qualitative analysis was described insufficiently. What kinds of the qualitative method was used? And how did the authors integrate the qualitative and quantitative results? These issues are critical for the quality of this research.

In the discussion, the first paragraph should contain a summary of the results.

The discussion should describe the research findings in international contexts. As the same as the background, this is a critical point for the publication of international journals.

Reviewer 2 Report

In the present manuscript, a breast oncology rotation for family residents was evaluated comparing rotation (RRs) versus non-rotation (NRRs) residents, using a pre-post knowledge questionnaire and semi-structured interviews.

Despite the sample size was small, this paper is interesting, and can be adapted to other training programs.

The manuscript can be accepted for publication if the authors are ready to incorporate the following revisions:

Introduction

In the introduction, the authors should better explain the importance of genetic testing to be suggested to breast cancer patients, as mutations in several susceptibility genes could emerge that could have important implications in both treatment and therapy.

In this regard, the authors would benefit from reading the following article, the contents of which could be useful for improving the manuscript:

PMID: 23940062 DOI: 10.1515/cclm-2013-0263

Materials and Methods

The authors do not clearly indicate how many FMs, both RRs and NRRs, initially were recruited, they should explain it better.

Furthermore, the terms ‘t1’ and ‘t2’ are indicative, but it must be indicated what they refer to when quoted for the first time.

Results

The ‘Demographics’section should be moved in ‘Materials and Methods’ section.

Equally, the ‘Rotation Structure and Content’ section should be moved in ‘Materials and Methods’ section.

Discussion

The ‘Summary of Findings’ section could be moved at the end of ‘Materials and Methods’ section.

Author Response

Please see attached, thank you.

Round 2

Reviewer 1 Report

The manuscript has been considerably improved. I think that this paper is suited for inclusion in our journal.